# Non-Thermal Plasma Application in Tumor-Bearing Mice Induces Increase of Serum HMGB1

**DOI:** 10.3390/ijms21145128

**Published:** 2020-07-20

**Authors:** Olga Troitskaya, Ekaterina Golubitskaya, Mikhail Biryukov, Mikhail Varlamov, Pavel Gugin, Elena Milakhina, Vladimir Richter, Irina Schweigert, Dmitry Zakrevsky, Olga Koval

**Affiliations:** 1Institute of Chemical Biology and Fundamental Medicine, Siberian Branch of the Russian Academy of Sciences, Akad. Lavrentiev Ave. 8, Novosibirsk 630090, Russia; troitskaya_olga@bk.ru (O.T.); sennuie@gmail.com (E.G.); biryukov.mm@ya.ru (M.B.); mvarlamov@gmail.com (M.V.); richter@niboch.nsc.ru (V.R.); 2Department of Natural Sciences, Novosibirsk State University, Pirogova Str. 1, Novosibirsk 630090, Russia; 3Rzhanov Institute of Semiconductor Physic, Siberian Branch of the Russian Academy of Sciences, Akad. Lavrentiev Ave. 13, Novosibirsk 630090, Russia; gugin@isp.nsc.ru (P.G.); lena.yelak@gmail.com (E.M.); zakrdm@isp.nsc.ru (D.Z.); 4Khristianovich Institute of Theoretical and Applied Mechanics, Siberian Branch of the Russian Academy of Sciences, Institutskaya Str. 4/1, Novosibirsk 630090, Russia; IVSchweigert@gmail.com

**Keywords:** HMGB1, non-thermal plasma, immunogenic cell death, anticancer treatment, proinflammatory cytokines

## Abstract

The application of cold atmospheric plasma (CAP) in cancer therapy could be one of the new anticancer strategies. In the current work, we used cold atmospheric plasma jet for the treatment of cultured cells and mice. We showed that CAP induced the death of MX−7 mouse rhabdomyosarcoma cells with the hallmarks of immunogenic cell death (ICD): calreticulin and heat shock protein 70 (HSP70) externalization and high-mobility group box 1 protein (HMGB1) release. The intensity of HMGB1 release after the CAP treatment correlated directly with the basal extracellular HMGB1 level. Releasing from dying cells, HMGB1 can act as a proinflammatory cytokine. Our in vivo study demonstrated that cold atmospheric plasma induces a short-term two-times increase in serum HMGB1 level only in tumor-bearing mice with no effect in healthy mice. These findings support our hypothesis that CAP-dependent HMGB1 release from dying cancer cells can change the serum HMGB1 level. At the same time, we showed a weak cytokine response to CAP irradiation in healthy mice that can characterize CAP as an immune-safety physical antitumor approach.

## 1. Introduction

While a great number of anticancer therapeutics have been developed, their efficiency is still limited resulting in a high rate of cancer-associated deaths. Besides chemotherapeutic treatment, physical approaches of tumor irradiation were also tested in various cancers. The application of cold atmospheric plasma (CAP) in cancer therapy solo or in combination with chemo- and immunotherapeutics could be one of the new anticancer strategies [1]. To reinforce CAP potential against cancers, it was combined with modern immunotherapeutic drugs, including checkpoint inhibitors, and this approach resulted in tumor suppression and prolonged survival [2]. Being generated at room temperature, CAP minimizes a risk of the thermal damage of irradiated tissue [3]. The generation of non-thermal atmospheric pressure plasma depends on the discharge geometry, the setup of operating gas or gas mixture flow and the methods of gas excitation. In general, the discharge device is a dielectric tube in which the flow of operating gas is organized and where the low temperature plasma is generated [4].

In the current work, we used a cold atmospheric plasma jet for the treatment of cultured cancer cells and mice. This approach has some advantages: the generation of ions, electrons and reactive oxygen (ROS) and nitrogen (RNS) species can be in the water and gas environment and their composition can be optimized by the investigators to stimulate various biological effects. ROS and RNS are highly reactive species and they function in routine biological processes as well as in tumor eradication treatment. Under CAP treatment, these compounds are believed to be the major inducers of cell death. However, the detailed molecular mechanism of the interference of CAP-generated ROS and RNS with cell death machinery remains elusive.

High mobility group box 1 protein (HMGB1) was initially identified as a DNA-binding non-histone chromatin protein which works as a DNA chaperon for the facilitation of the binding of transcription factors [5]. Later, another functional activity of HMGB1 was showed where it mediated the activation of innate immune responses in particular against dying cells [6]. Several anticancer drugs can induce an immunogenic type of cancer cells death (ICD) which results in the activation of an intrinsic immune response against tumor cells reinforcing their therapeutic efficiency. Immunogenic cell death is associated with calreticulin (CRT) translocation to the outer cellular membrane and the release of ATP and HMGB1 [7,8]. Recently, Lin and co-authors have demonstrated that dielectric barrier discharge (DBD)-generated CAP induced the ICD of cancer cells in vivo [9]. As a substantial contributor to ICD, HMGB1 activates the immune system by being released from dying tumor cells and prompts the cross-presentation of tumor-associating antigens through its ligation and triggering of TLR4 on dendritic cells (DC) [10]. For CAP activity, it was verified that the CAP treatment of B16F10 cells promoted DC maturation in vitro [2]. In addition, Yoon and co-authors have also shown that CAP induced the expression of HMGB1 in cancer cells [11]. These findings indicated that CAP can change the intra- and extra-cellular level of HMGB1.

In this work, our hypothesis was that CAP may also cause a shift of blood HMGB1 level in treated mice and activate cytokine responses.

## 2. Results

### 2.1. Cold Atmospheric Plasma Induces Death of Murine MX−7 Cancer Cells with ICD Markers

In our previous studies [12,13], we have shown that the direct irradiation of A549 and A431 human cancer cells by CAP in helium with U = 4.2 kV for 30–120 s reduced cell viability up to 25%. Here, we tested the CAP activity against mouse MX−7 rhabdomyosarcoma cells under conditions that were close to these conditions. The experimental CAP device is a quartz tube with the powered electrode and capillary inside which operates with a frequency of 40 kHz and a voltage amplitude of 2–10 kV. The plasma jet consists in a sequence of streamers which propagate from the powered electrode to the target over the laminar inert gas flow.

Figure 1 demonstrates the parameters of the CAP setup. The applied voltage, discharge current and a ROS/RNS part of the spectrum of the plasma jet are show for the working gas helium. As seen in Figure 1b, the streamer is generated during a positive phase of the voltage and indicated with a peak of the discharge current.

MX−7 cells were directly irradiated by CAP for 5–120 s and 24 h after the MTT test was performed to analyze the cell viability (Figure 2a). After the plasma irradiation, the cells were grown in complete media in all experiments.

The highest effect was achieved for 60 s of irradiation and this further increase in the irradiation period up to 120 s did not lead to a decrease in cell viability. The proliferation of the real-time curves of CAP-irradiated cells indicated that long treatment (2 and 4 min) had a strong killing effect which occurred immediately after the CAP exposure (Figure 2b). Despite the fact that the jet temperature did not exceed 40 degrees, the duration of treatment increases the risk of the thermal damage of cells and damage caused by the associated UV radiation. Therefore, the correction of the exposure duration is of great importance. One-minute treatment suppressed the cell proliferation, and combined with the MTT data, these results allow us to use one-minute treatment as an optimal condition of irradiation in the subsequent experiments. The microscopic analysis of dying cells after CAP treatment revealed the appearance of cells with membrane blebbing or apoptotic bodies that are characteristic of apoptosis (Figure 2c). To confirm apoptosis-like cell death, phosphatidylserine externalization and nucleus integrity were analyzed by flow cytometry using Annexin V/propidium iodide (PI) staining. CAP irradiation produced a substantial population of cells with Annexin V positive phenotype as well as cells in late apoptosis or necrosis: double positive Annexin V/PI cells (Figure 2d,e). Thereby, these data demonstrated that 24 h after the CAP irradiation, the cellular morphology was changed to an apoptosis-related phenotype with phosphatidylserine externalization, that confirmed the involvement of apoptosis in CAP-induced cell death.

The calreticulin and heat shock protein 70 externalization and the release of HMGB1 and ATP are the main hallmarks of immunogenic cell death (Figure 3a) [8]. To find ICD hallmarks in CAP-irradiated cells (4.9 kV, 1 min), we analyzed the extracellular HMGB1 as well as the translocation of CRT and HSP70 to the outer cellular membrane. Extracellular HMGB1 was estimated by ELISA as described in Methods. The increase in HMGB1 was detected 24 and 32 h after the CAP irradiation (Figure 3b). To verify the intensity of the HMGB1 release, we tested the extracellular HMGB1 in other cancer cell lines—murine colorectal cancer cells CT26 and human epidermoid carcinoma cells A431. The highest concentration of HMGB1 was detected in the culture medium of CT26 cells 24 h after the CAP treatment. We noticed that the basal HMGB1 concentrations in the non-treated cells were also different for all three tested cell lines. To estimate the relative changes of HMGB1, the data were normalized to the non-treated cells (Figure 3c). The maximal relative increase in HMGB1 was shown in the MX−7 cells with the lowest basal HMGB1 level in the non-treated cells. It was also confirmed by the Western blot analysis of total cellular HMGB1 when we found a decrease in cellular HMGB1 (Figure 3d). We can conclude that CAP-stimulated HMGB1 release from treated cells depends on the basal HMGB1 level. Our finding that HMGB1 extracellular concentration was higher for the 24 h after the irradiation than 32 h after for all the tested cell lines, as a result of the time-dependent proteolysis of HMGB1 by the proteolytic enzymes from dying cells. Such digested forms of HMGB1 can create difficulties for ELISA analysis.

The externalization of cellular proteins may be detected by the staining of the cells with some appropriate antibodies with no fixation and permeabilization steps (Figure 4a). The staining of fixed and permeable cells allows us to detect the total cellular protein. The externalization of calreticulin (ecto-CRT) and HSP70 (ecto-HSP70) was measured by flow cytometry 6 and 24 h after the CAP treatment (Figure 4b). Ecto-CRT-positive cells were observed only 24 h after the plasma exposure (Figure 4 c). Finally, ecto-HSP70 was also tested 24 h after the CAP treatment and up to 38 % cells were ecto-HSP70-positive (Figure 4d). There were no differences in CRT- or HSP70-positive cells in the control and plasma-treated cells when the cells were fixed and permeabilized (Figure 4e,f), which confirmed the translocation of these ICD markers to the outer cellular membrane in the CAP-irradiated cells.

Thus, we can conclude that 24 h after the CAP-treatment, the ICD markers were found. The detection of a specific set of ICD hallmarks in the treated cells means that the CAP irradiation leads to an immunogenic type of cell death.

### 2.2. Cold Atmospheric Plasma Induces the Increase in HMGB1 in Blood Serum of Tumor-Bearing Mice

Theoretically, extracellular HMGB1 from dying cells can reach the bloodstream. We then evaluated whether CAP shifts the HMGB1 blood serum level in irradiated mice. We hypothesized that CAP-dependent HMGB1 release from dying cancer cells can change the serum HMGB1 level. C3H/He mice were subcutaneously injected with MX−7 cells (1.0 × 10^6^ cells in 0.1 mL PBS per animal). Healthy mice and tumor-bearing C3H/He mice with an average tumor size 50 mm^3^ received two courses of CAP irradiation as described in Figure 5a and Methods section. The mice were immobilized by anesthesia to apply the standard treatment, and the zone of irradiation of the tumor-bearing mice was the same for the healthy mice (see Figure 5b). The parameters of the CAP were selected based on the in vitro studies. To validate that CAP treatment does not cause thermal damage to mice, we used a thermal visor to measure the skin temperature in the treated target. The median skin temperature detected over the time course of the irradiation was 37.8 ± 0.7 °C and no thermal damages were observed. The mouse blood was collected 1 h and 24 h after the second CAP treatment. The analysis of mice serum revealed that the CAP treatment induced a two-time increase in HMGB1 1 h after the second CAP exposure in the tumor-bearing mice (Figure 5c). It should be mentioned that 24 h after the second CAP irradiation, the HMGB1 level in the serum decreased in both CAP-treated groups. We assumed that no elevation in the serum HMGB1 in healthy mice exposed to CAP indicates the safety of CAP irradiation for healthy tissue.

### 2.3. Cold Atmospheric Plasma Changes Level of Serum Pro-Inflammatory Cytokines

The elevated level of cytokine secretion can also be a marker of an arising immune response. We analyzed the serum level of IL−1α, Il−1β, IL−4, IL−6, IL−10, IL−12, IL−17α, INF- γ, TNF- α, G-CSF and GM-CSF in healthy mice after CAP irradiation. Difference in the cytokine level between the control mice and the CAP-treated mice was detected only for two cytokines: G-CSF significantly increased and IL−4 decreased in the serum of irradiated healthy mice (Figure 6).

Taking into account the data of the HMGB1 serum level in healthy irradiated mice, our finding shows that short-term CAP irradiation induces a weak cytokine response.

## 3. Discussion

The mechanism of CAP interaction with the immune system is poorly understood. Plasma contains the cocktail of different radicals and ions, reactive oxygen and nitrogen species, and ultraviolet which can react with irradiated tissue, as well as an electric field [14]. On the cellular level, ROS and RNS have various effects in signal transduction; their excess can result in oxidative damage, cell death, and various diseases. An essential increase in ROS in mice tumors was registered in vivo after plasma jet treatment [15,16]. CAP is a cold plasma with no thermal damage of the treated bio-surface; therefore, it is likely that ROS and RNS are the major mediators in CAP interaction with the immune system—direct or indirect. Molecular signaling, including the H_2_O_2_-dependent activation of MAPK and NF-κB pathways, may be induced by CAP irradiation in immune and non-immune cells.

The potential selectivity of CAP towards cancer cells has been demonstrated in various cell lines [17]. Such selectivity to cold atmospheric plasma treatment can depend on cancer cell metabolism. Usually, rapidly developing tumors are characterized by the lag development of blood microvessels and exist in low oxygen conditions known as ‘hypoxia’ [18,19]. The imbalance between oxygen supply and consumption in the tumor microenvironment is overcome by switching the cancer cell metabolism to glycolysis. Launching the ‘hypoxia’ leads to changes in the tumor cell proteome as well as genomic instability [18].

The cellular redox environment regulates the temporal functions of HMGB1. HMGB1 (previously HMG1; HMG−1; HMG 1; amphoterin; p30) contains three cysteine residues—C023, C45 and C106—which can form disulfide bonds [20]. Nuclear HMGB1 is completely reduced while disulfide bond-containing HMGB1 functions as a proinflammatory cytokine [21], the same role it can also play when secreted by immune cells. It is possible that CAP irradiation with ROS and RNS results in the direct oxidation of nuclear HMGB1 and stimulates its release in a reduced form. Here we demonstrated that under these conditions, CAP irradiation leads to cell death with HMGB1 release from the dying cells. Extracellular HMGB1 functions as an immune adjuvant to trigger a robust response to the activation or suppression of T cells, dendritic cells, and endothelial cells; it is also involved in the regulation of innate and adaptive immune responses [22,23,24,25]. Tumor antigens from dying cells can activate antitumor adaptive immune response, where HMGB1 plays a critical role (Figure 3 a). Moreover, HMGB1 acts as an adjuvant for several vaccines [26,27]. Kang and co-authors describe the increasing evidence from in vitro and in vivo studies that HMGB1 is a critical regulator of DC maturation and function [25]. Releasing from dying cells, HMGB1 promotes DC maturation [28] and HMGB1/TLR4 interaction is required for the DC-mediated antitumor immune response in conventional anticancer therapies, such as chemotherapy and irradiation [29].

HMGB1 release is a hallmark of the immunogenic type of cell death, therefore other markers of ICD were also analyzed in CAP-irradiated cells. ICD is a form of regulated cell death that is sufficient to activate an adaptive immune response in immunocompetent syngeneic hosts [8]. Various clinically used anticancer drugs, including doxorubicin and oxaliplatin were demonstrated to induce the ICD of cancer cells [7,29,30,31]. We showed that the CAP treatment of MX−7 cells stimulated calreticulin and HSP70 externalization. It should be mentioned that the time slots of ICD hallmarks in plasma-treated cells were different from conventional ICD inducers [32]. For example, ecto-CRT can be detected 1–4 h after the doxorubicin treatment, while here we did not found ecto-CRT 6 h after the CAP irradiation. In addition, the plasma investigators who described the ATP release from plasma-treated cells in fact detected the ATP release 10 min after the treatment [9,33]. In contrast, conventional ICD inducers stimulate ATP release 18 h after the drugs’ addition [32]. These divergences point out the problem: would the time-shifted manifestation of ICD hallmarks in plasma-treated cells change the interaction between these molecules and immune cells? We believe that the indicated problem has to be solved in the near future.

HMGB1 interacts with various receptors, including RAGE, TLRs (−2, 4, 9), integrin Mac−1, α-synuclein, CD24, and cytokines [25]. The multiplicity of binding partners makes the unambiguous interpretation of HMGB1 function in cancer difficult at the moment. HMGB1 protein and RAGE are identified as a ligand–receptor pair that plays an important role in regulating the invasiveness of tumor cells. Since extracellular HMGB1 can act as a proinflammatory cytokine, we tested blood serum HMGB1 in healthy and tumor-bearing mice after the CAP treatment. Here, we showed an increase in serum HMGB1 after the CAP irradiation with no elevation in serum HMGB1 in healthy mice exposed to CAP. It is likely that the CAP irradiation of healthy tissues does not locally activate proinflammatory reactions. In contrast, HMGB1 release from dying cancer cells can activate anticancer immunity: the complex of HMGB1 with RAGE or Toll-like receptor 4 (TLR4) triggers cascades of reaction that lead to the local secretion of proinflammatory cytokines, including IL−1, IL−6 and TNF-α [34].

To check the immunosafety of CAP irradiation for the animals, we analyzed the blood cytokine level in healthy mice. We observed solely the increase in granulocyte colony stimulating factor (G-CSF) in mice serums. G-CSF belongs to the family of colony stimulated factors (CSF) and they function to stimulate granulopoiesis and innate immunity [35]. The chronic increase in serum G-CSF can result in inflammatory and/or autoimmune states. Solely the stimulation of G-CSF with no difference in other proinflammatory cytokines, including INFy and TNFa, enable us to determine the CAP as an approach with weak influence on cytokines. Overall, further in-depth study of the interaction between CAP irradiation and the immune system will make it possible to assess the immune safety of this method for its translation to the clinic.

## 4. Materials and Methods

### 4.1. Materials, Cell Lines and Animals

The following antibodies and chemicals were obtained from commercial sources: rabbit ant-h/m/rat CRT (Abcam, ab2907), mouse ant-h/rat HSP70 (RnD, 841680), Alexa Fluor 594-conjugated chicken anti-rabbit antibodies (Invitrogen, Belgium), Alexa Fluor 488-conjugated goat anti-mouse antibodies (Invitrogen, Belgium), polyclonal rabbit-anti-mouse and mouse-anti-rabbit HRP-conjugated antibodies (Biosan, Russia), Triple-Express (GIBCO, Thermo Fisher, NY, USA), doxorubicin (Teva Pharmachemie B.V., Netherlands).

MX−7 murine rhabdomyosarcoma cells were obtained from the Russian cell culture collection (Russian Branch of the ETCS, St. Petersburg, Russia). CT26 murine colorectal cancer cells were kindly provided by Dr. Nadezhda Gurskaya (M. M. Shemyakin and Yu. A. Ovchinnikov Institute of Bioorganic Chemistry, Moscow, Russia). A431 human epidermoid carcinoma cells were purchased from ATCC (CRL−1555). Cells were grown in Dulbecco’s modified Eagle’s medium (DMEM, Sigma-Aldrich) supplemented with 10% fetal bovine serum (GIBCO, Thermo Fisher Scientific, Waltham, MA, USA), 2 mM L-glutamine, 250 mg/mL amphotericin B and 100 U/mL penicillin/streptomycin. Cells were maintained as previously described [36]. Cells were free of mycoplasma or ureaplasma (Appendix A).

Female C3H/He mice (6–8 weeks old) were obtained from the super pathogen free (SPF) vivarium of the Institute of Cytology and Genetics SB RAS, Novosibirsk, Russia.

### 4.2. Cold Atmospheric Plasma System and Treatment Parameters

The plasma source of a cold plasma jet described in [13], the design of which is similar to the device presented in [37], was used in this study. The plasma source was a quartz discharge channel with the powered electrode inside and external ring grounded electrode over the dielectric tube. A sinusoidal voltage signal with a frequency of ~ 40 kHz and an amplitude of up to 10 kV was applied to the electrodes. A dielectric capillary with a 2.3 mm diameter was placed into the excitation volume. When the voltage was applied to the electrodes and the helium flow rate was 1–10 L/min, the discharge was ignited. The streamers form near the powered electrode and propagate inside and outside of the plasma device composing a visually uniform plasma jet. Typical discharge parameter values were as follows: the gas flow rate v = 9 L/min, voltage amplitude *U* = 4.9 kV, discharge current *I* was up to 12 mA, current half-width duration 15 ns, pulse energy was up to 12 μJ.

### 4.3. Cell Viability Assays

Cell viability was detected 24 h after the direct plasma irradiation using the MTT test and real-time iCELLigence system (ACEA Biosciences, iCelligence system, ASEA Biosciences, South Korea) as was described previously [36].

Bright-field microscopy was made using the ZOE™ Fluorescent Cell Imager (BioRad Laboratories).

### 4.4. Flow Cytometry Analysis

All the tests were performed using FACSCanto II flow cytometer (BD Biosciences, Franklin Lakes, NJ, USA), and the data were analyzed by FACSDiva Sofware (BD Biosciences). Cultured cells were initially gated (P1) based on forward scatter versus side scatter to exclude small debris, and ten thousand events from this population were collected. Annexin V/PI staining was performed as was described previously [36].

For ecto-CRT or ecto-HSP70 detection, the cells were incubated with primary anti-CRT antibodies (1:100), or anti-HSP70 antibodies (1:100) for 1 h at 23 °C. Rabbit IgG (Thermo Fisher Scientifc, Waltham, MA, USA) or mouse IgG (R&D, USA) was used as an isotype control. The cells were initially gated based on forward versus side scatter to exclude small debris (Appendix A). For the total CRT and HSP70 analysis, the cells were incubated with 10% formalin and 0.1% Triton X100 in PBS for fixation and permeabilization.

### 4.5. HMGB1 Detection

For the quantitative determination of HMGB1 in the culture medium or in mice serum, enzyme immunoassays (IBL International, Hamburg, Germany) were used. The samples were prepared in according to the manufacturer’s protocols and their optical density was detected with a multichannel spectrophotometer Apollo LB912 (Berthold Technologies, TN, USA) at 450 nm (reference wavelength 620 nm). The total cellular HMGB1 was detected by Western blot as was described previously [38].

### 4.6. Cytokines ELISA Assay

Serum cytokines were measured using the Multi-Analyte ELISAArray Kit MEM−004A (Qiagen, USA) where the level of IL−1α, Il−1β, IL−4, IL−6, IL−10, IL−12, IL−17α, INF- γ, TNF- α, G-CSF and GM-CSF can be assessed. Murine blood was isolated from the mice 1 h after the second CAP irradiation, and the serums were prepared as described previously [38]. Serum samples were diluted 8 times for subsequent analysis. The diluted samples were loaded into a 96-well ELISA microplate and analyzed according to the manufacturer’s instruction. CAP-treated cytokine values were normalized to non-treated values (100%) and presented as the relative (% from control) amount.

### 4.7. Mice Manipulations and Treatment

MX−7 rhabdhomiosarcoma cells growing in a 25 mm^2^ culture were collected with trypsin, washed once in PBS, and then resuspended in PBS for inoculation in C3H/He mice. C3H/He mice were inoculated subcutaneously with 5 × 10^5^ MX−7 cells.

The mice were anesthetized by the injections of TBE solution (0.2% 2,2,2-Tribromoethanol, 0.2% 2-Methylbutanol in 0.9 % NaCl). TBE (400 μL) was injected intraperitoneally in mice, and 5 min later the mice were asleep. Tumors or points for the CAP irradiation in healthy mice were marked by pen and the animals were put on a table so that the marked points were 2 cm under the plasma jet. CAP were exposed for 1 min with voltage U = 4.9 kV, current I = 7 mA, current half-width duration 15 ns and pulse energy ~7 μJ. For the temperature measurements, a thermal visor Testo 872 (Testo AG, Germany) was placed above the treated mouse.

### 4.8. Ethic Statement

All the animal experiments were carried out in compliance with the protocols and recommendations for the proper use and care of laboratory animals (EC Directive 86/609/EEC for animal experiments). The protocols were approved by the Committee on the Ethics of Animal Experiments of the Administration of the Siberian Branch of the Russian Academy of Science (Protocol Number 41, 4 April 2018). The mice were housed under super pathogen free (SPF) conditions in ventilated animal cabinets under controlled lighting conditions at 65% humidity, 25 °C, with 10/14 h light–dark cycle and allowed food and water ad libitum. The animals were euthanized by exposure to CO2.

### 4.9. Statistics

Significance was determined using a two-tailed Student’s *t*-test. A *p* value of less than 0.05 was considered significant. All the error bars represent the standard error of the mean.

## Figures and Tables

**Figure 1 ijms-21-05128-f001:**
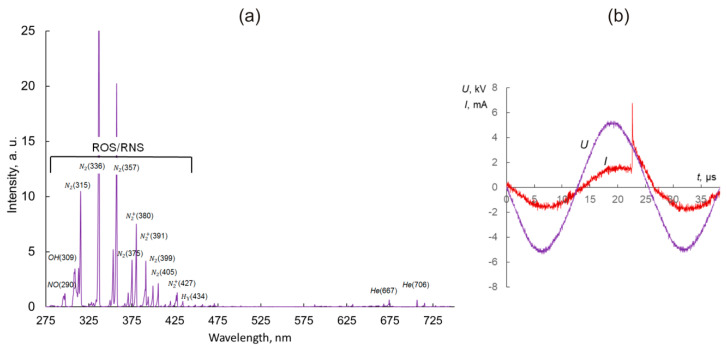
Characterization of the cold atmospheric plasma (CAP) setup. (**a**) Optical emission spectroscopy (OES) spectrum of the CAP in helium, v = 6 L/min, U = 4.9 kV. (**b**) Typical discharge voltage, frequency, and wave form for the CAP device for U = 4.9 kV.

**Figure 2 ijms-21-05128-f002:**
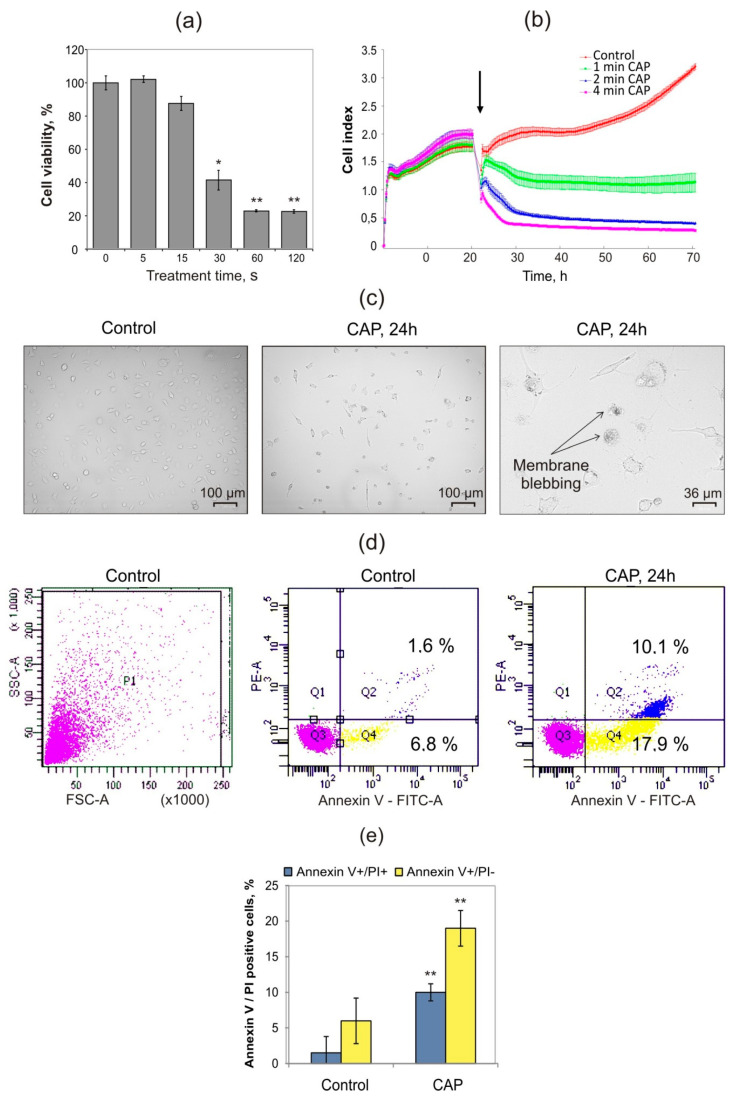
Plasma induces the death of MX−7 cells. (**a**) Cell viability (MTT) assay was performed 24 h after the CAP treatment. The data are presented as a percent of the non-treated cells (100%). (**b**) Real-time iCELLigence curves for the MX−7 cells exposed to CAP. The black arrow indicates the moment of irradiation; (**c**) Bright-field microscopy of the CAP-treated MX−7 cells. Representative photos of three independent repeats. (**d**) Flow cytometry analysis of apoptosis in the CAP-treated MX−7 cells. Cells were stained with Annexin-V/propidium iodide (PI). Ten thousand cells were recorded in the P1 population where Annexin-V + /PI- cells then were defined as apoptotic cells, Annexin-V + /PI + were defined as late apoptotic/secondary necrotic cells. Representative histograms of three independent repeats. (**e**) Quantification data for Annnexin V/PI positive cells. Student’s *t*-test was applied to compare the control and experimental groups. Data are represented as the mean ± SEM, * *p* ≤ 0.05 and ** *p* ≤ 0.01.

**Figure 3 ijms-21-05128-f003:**
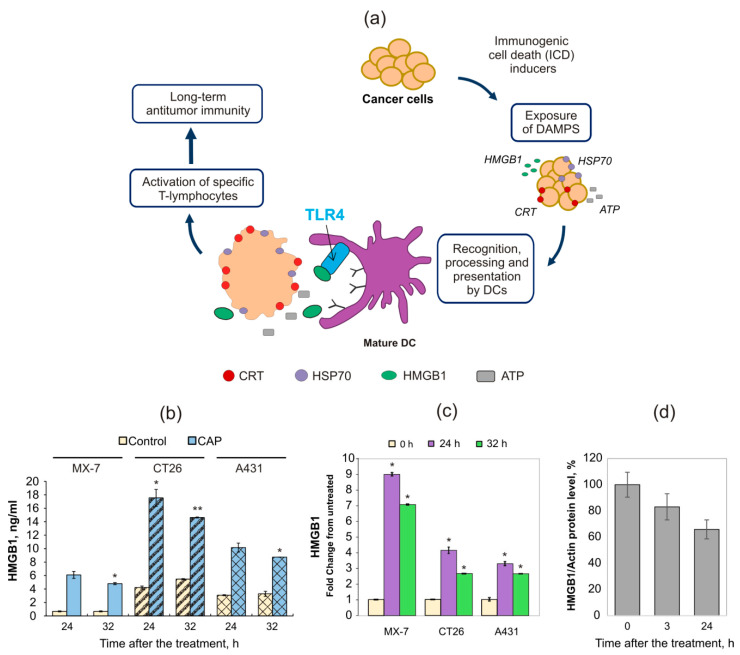
Plasma induces the death of irradiated cells with the appearance of immunogenic cell death (ICD) hallmarks. (**a**) The scheme of ICD-related molecular transformations and cellular signaling. DCs—dendritic cells; DAMPS—danger associated molecular patterns. (**b**–**d**) Analysis of extracellular (**b**,**c**) and intracellular (d) HMGB1. MX−7, CT26 and A431 cells were exposed to CAP for 1 min, and 24 as well as 32 h later HMGB1 was assayed in the culture medium. (**c**) Data are presented as the fold change in the HMGB1 amount compared to the non-treated control. (**d**) Cellular HMGB1 in the CAP-treated samples; relative quantification of HMGB1/α-actin according to the Western blot analysis of the HMGB1 expression in the MX-7 cell lysates. Student’s *t*-test was applied to compare the control and the experimental groups. Data are represented as the mean ± SEM, * *p* ≤ 0.05, ** *p* ≤ 0.01.

**Figure 4 ijms-21-05128-f004:**
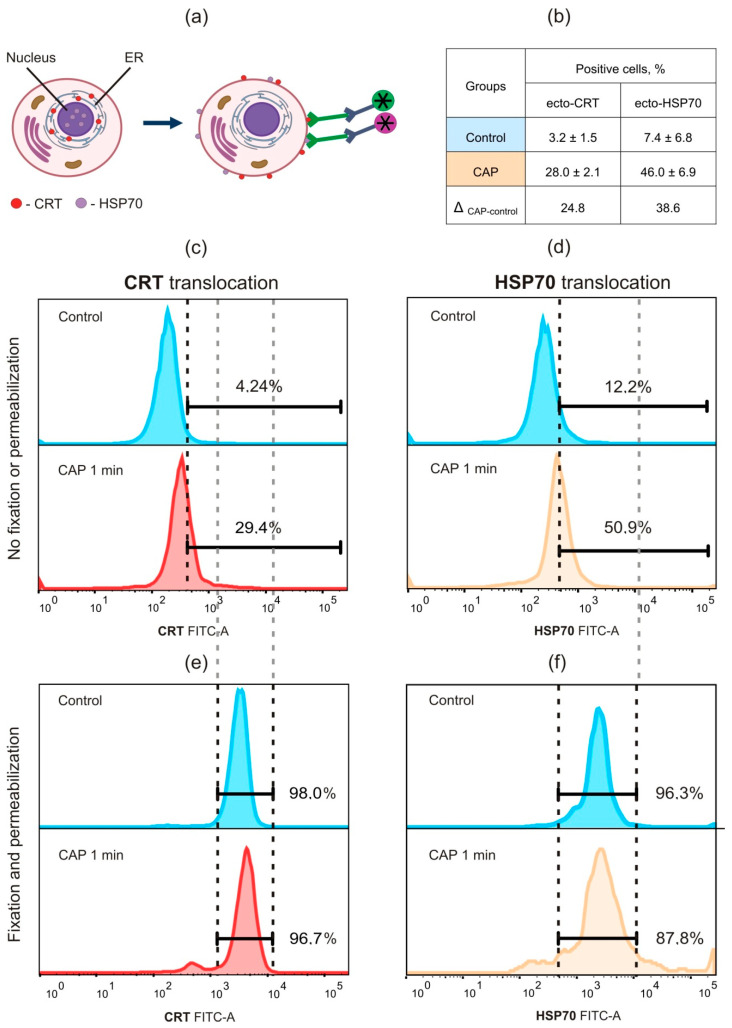
CAP-dependent translocation of CRT and HSP70. (**a**) The scheme of the analysis of CRT and HSP−70 translocation to the outer cellular membrane. MX−7 cells were irradiated by CAP for 1 min and 24 h after ecto-CRT and ecto-HSP70 were analyzed. (**b**) Quantification data for the ecto-CRT and ecto-HSP70 positive cells. (**c**–**f**) Analysis of the extracellular (**c**,**d**) and total (**e**,**f**) CRT and HSP70. MX−7 cells were stained with anti-CRT (**c**,**e**) and anti-HMGB1 (**d**,**f**). Staining was performed with (**e**,**f**) or no (**c**,**d**) fixation and permeabilization. Representative graphs of three independent repeats.

**Figure 5 ijms-21-05128-f005:**
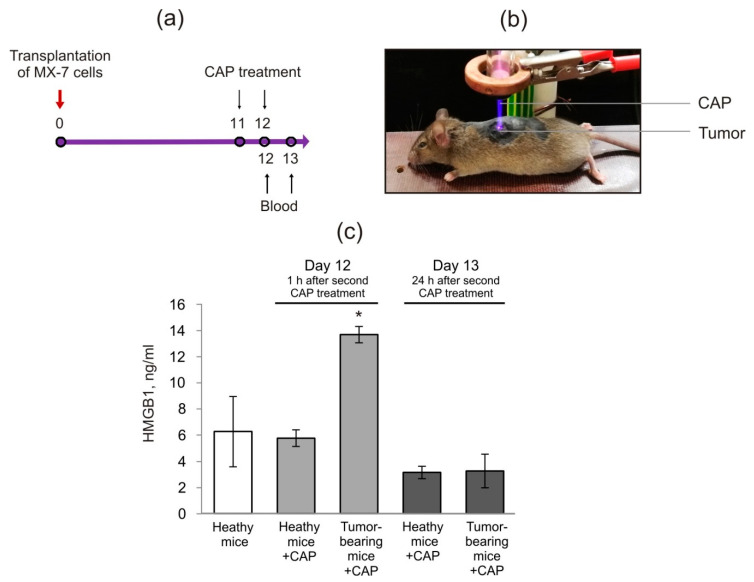
The influence of CAP irradiation on the serum HMGB1 in healthy and tumor-bearing mice. (**a**) Scheme of the experiment. C3H/He mice were inoculated subcutaneously with 5*10^5^ MX−7 cells. (**b**) Image of mouse irradiation. (**c**) Serum HMGB1 amount. Data are presented as the mean ± SEM (*n* = 5). The differences were calculated using the Student’s *t*-test and these were significant with *p* < 0.05 (*).

**Figure 6 ijms-21-05128-f006:**
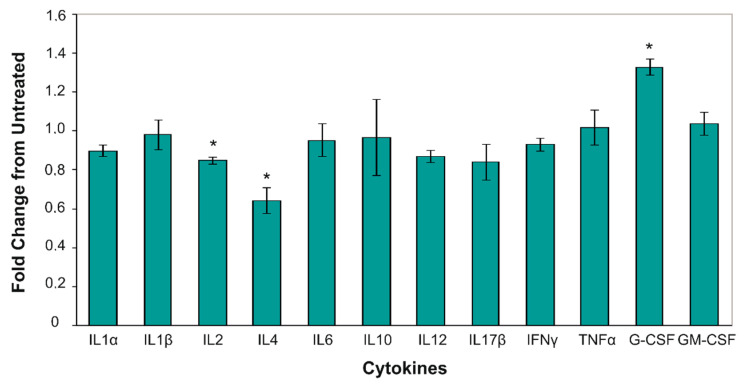
Cytokine levels in the blood serum of CAP-treated mice. The relative blood cytokine level in the CAP-irradiated mice serum. Data were normalized to the non-treated control and presented as the mean ± SEM (*n* = 5). Blood samples were collected on day 12, 1 h after the second CAP irradiation. The differences were calculated using the Student’s *t*-test, the difference between groups was statistically significant at *p* < 0.05 (*).

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
