# Peer review of "Non-Thermal Plasma Application in Tumor-Bearing Mice Induces Increase of Serum HMGB1"

_ijms, 2020, doi:10.3390/ijms21145128_

Round 1

Reviewer 1 Report

In the revised manuscript by Troitskaya et al., the data is better presented than the previous version. However, there are certain issues that need to be addressed to make the manuscript better readable to potential audience.
Overall the authors have done a good job revising the manuscript. These concerns need to be taken into consideration before accepting the manuscript for publication:

  1. Line 100-102, the authors mention about jet temperature. What exactly was the jet temperature during the present study. Can the authors include this information in the manuscript.
  2. Fig.2C How many times the experiment was repeated. Quantification data for Annnexin V/PI positive cells required.
  3. Similarly for Figure 4, Quantification data is required. The plots are difficult to comprehend. Having the data quantified and showing the representative plots may be a better way of representation of data.
  4. Figure 6. Instead of fold change, can the authors present the actual numbers. A better way to represent this data is to present a table form, where you have two columns of: 1)control and 2)treated and the values of cytokines as Mean±SEM is presented. This way we know what were the actual levels of the cytokines in pg/ml and will help other researchers working in the area get a better picture of the anticipated changes in the levels of cytokines in vivo after treatment.

Author Response

Reviewer 1

In the revised manuscript by Troitskaya et al., the data is better presented than the previous version. However, there are certain issues that need to be addressed to make the manuscript better readable to potential audience.

Overall the authors have done a good job revising the manuscript. These concerns need to be taken into consideration before accepting the manuscript for publication:

Answer

We thank the reviewer for the positive feedback on our manuscript.

Comment 1   

Line 100-102, the authors mention about jet temperature. What exactly was the jet temperature during the present study. Can the authors include this information in the manuscript.

Answer 1

We included this information (see Lines 180-183) and measurement details (Lines 353-354) in the manuscript. Investigations of the temperature of a plasma jet have been carried out in a number of works (see, for example, Laroussi M. 2009 Low-temperature plasmas for medicine. IEEE Trans. Plasma Sci. 37 714–25; Reuter S. et al. J. Phys. D: Appl. Phys. 2018; 51: 233001).

In this study, it was important how much the temperature of a biological object rises when exposed to a plasma jet. To validate that CAP treatment does not cause thermal damage to mice, we used a thermal visor to measure skin temperature in the treated target. Median skin temperature detected over the time course of the irradiation was 37.8 ± 0.7 °C and no thermal damages were observed. For temperature measurements, a thermal visor Testo 872 (Testo AG, Germa, ny) was placed above treated mouse.

Comment 2   

            Fig.2C How many times the experiment was repeated. Quantification data for Annnexin V/PI positive cells required.

Answer 2

We included quantification data for Annexin V/PI positive cells (see Fig. 2f). Experiments (Fig. 2c and 2d) were repeated at least three times.

Comment 3

            Similarly for Figure 4, Quantification data is required. The plots are difficult to comprehend. Having the data quantified and showing the representative plots may be a better way of representation of data.

Answer 3

We included quantification data for ecto-CRT and ecto-HSP70 positive cells in the table (see Fig. 4b).

            Comment 4

            Figure 6. Instead of fold change, can the authors present the actual numbers. A better way to represent this data is to present a table form, where you have two columns of: 1)control and 2)treated and the values of cytokines as Mean±SEM is presented. This way we know what were the actual levels of the cytokines in pg/ml and will help other researchers working in the area get a better picture of the anticipated changes in the levels of cytokines in vivo after treatment.

Answer 4

            We totally agree with the comments on the figure 6. Unfortunally, the kit (Multi-Analyte ELISAArray Kit MEM-004A (Qiagen, USA) https://www.b2b-qiagen.com/be/shop/protein-and-cell-assays/multi-analyte-elisarray-kits?catno=MEH-006A#productdetails) does not come with a full standard curve; all we have is a single well that corresponds to one cytokine (positive cytokine control). So, we triplicated each blood sample to obtain mean values and to calculate SEM and obtained only one value for positive cytokine control. The values we get from this kit are meant to be relative, not absolute. This kit is a good way for the pilot screening investigation: the idea is to profile a range of cytokines (whether present/absent and whether more/less depending on what you are comparing it to) to identify the ones which are modulated in your experimental setup. Once we identify the more interesting ones, we are supposed to run single analyte ELISAs with full standard curves for accurate quantification with another commercial kits with established standard curves. We realized that this multi-analyte kit is just a good starting point, not the whole picture. We believe that in next experiments we can focus our investigation on the cytokines which demonstrated statistically significant changes in this work – IL-2, IL-4 and G-CSF. Only using kits with full standard curve we can get concentration values from the OD readings.

            As such, we request to reviewer to stay Fig.6 data in relative fold change of cytokines.

Reviewer 2 Report

N/A

Author Response

We thank the reviewer for the positive feedback on our manuscript.

Round 2

Reviewer 1 Report

Fig.2f. Quantification data for Annexin V/PI. Please do a double check. Should the yellow bars be AnnexinV+/PI-  instead of Annexin V-/PI+ as demonstrated by the authors. The X-axis values are clearly Annexin V+/PI- cells and not Annexin V-/PI+ cells. The labels on the figure need to be corrected.

Author Response

Reviewer 1

Fig.2f. Quantification data for Annexin V/PI. Please do a double check. Should the yellow bars be AnnexinV+/PI-  instead of Annexin V-/PI+ as demonstrated by the authors. The X-axis values are clearly Annexin V+/PI- cells and not Annexin V-/PI+ cells. The labels on the figure need to be corrected.

Answer

We totally agree with the comments on the figure 2f. It was a typing error which is corrected now (See Fig. 2f).

This manuscript is a resubmission of an earlier submission. The following is a list of the peer review reports and author responses from that submission.

Round 1

Reviewer 1 Report

Troitskaya et al present a study on the use of non-thermal plasma/cold atmospheric plasma (CAP) in tumor-bearing mice and check the level of immunogenic cell death (ICD) biomarkers like HMGB1, Calreticulin and HSP70 level in blood serum. To confirm the efficacy of CAP to induce ICD, cell viability was measured at different time intervals using MX-7 cell lines (Murine rhabdomyosarcoma), CT26 (Murine colorectal cancer cells) and A431 (Human epidermoid carcinoma cells). Afterwards, MX-7 cell was used to measure expression levels of Calreticulin and HSP-70 in in vitro and in vivo studies. The different cytokine levels were measured using blood serum. The study is very basic and the authors need to perform a number of additional experiments to confirm the results. Overall, the incomplete results and need to add experiments according to the available literature makes it unsuitable for publication at this stage. However, the comments appended below makes it better for the future consideration in another journal.

  1. As the literature suggested, HMGB1 contributes to increase cancer pathogenesis and after its extracellular release it combines with the RAGE protein which is the main signaling pathways for the initiation of the cancer. The authors need to clarify how CAP facilitates decreased cancer cell viability?
  2. In Figure 2g and 2h the dot-plot showed the expression of HSP70. It is clearly visible that there are two different cell populations present. The authors should check mycoplasma tests or perhaps another DIC microscopy image check to cross confirm if it is due to cross contamination of different cells? It is strongly recommended to provide a graph between SSC vs FSC to eliminate the chance of dual population or to know where the problem exists.
  3. The result shows the level of HMGB1 reduces after 24h in all 3 cell lines. It would be interesting to know if the Calreticulin and HSP70 also follow the same pattern of regression or not. The authors should consider taking a 32h time point also while measuring the above 2 expression levels.
  4. The authors must increase the voltage or current of the flow cytometry to change the histogram peak from extreme left to the middle. Figure 2e and 2f peak representation must be changed as suggested. Also, the peak shift measurement should be done separately by using appropriate software and make a graph to represent it.
  5. There are no statistical error bars in figure 2d.
  6. As stated, the mice blood was withdrawn the 2nd day after CAP treatment. According to Figure 3a, CAP treatment was performed on the 11th and 12th days and blood serum was isolated on the 12th and 13th days, respectively. At what day (12/13 day) were the HMGB1 expression and cytokines expression levels checked? Also, the same figure mentions that the experiment was run for up to day 20 but the serum was collected on the 12th and 13th Is there any specific reason to end the experiment at the 20th day?
  7. The author must mention about each peak in the OES graph in figure 1b and consider the x-axis only up to 475nm.

Reviewer 2 Report

In  vitro, the authors have observed that CAP-treated cancer cells show hallmarks of immunogenic cell death. However, analysis of cytokine levels in blood serum of CAP-treated mice show that there was non-significant increase in the levels of pro-inflammatory cytokines such as TNF-alpha, IL-6, Il-1beta etc.  The results seem contrasting as induction of immunogenic cell death is expected to increase in the levels of inflammatory cytokines and chemokines and subsequently lead to activation of immune cells. The manuscript fails to describe how CAP does not induce pro-inflammatory response despite being an inducer of ICD. Is the mechanism known?

Page 8  Line 249-252, Cell viability assay: Were the cells grown in serum free media or complete media after plasma irradiation?

Page 8,  Line 275-276: Serum samples were dissolved eight times for subsequent analysis? Do the authors mean diluted “8-times”